# Composite Polymer for Hybrid Activity Protective Panel in Microwave Generation of Composite Polytetrafluoroethylene -Rapana Thomasiana

**DOI:** 10.3390/polym13152432

**Published:** 2021-07-23

**Authors:** Ionel Dănuț Savu, Daniela Tarniță, Sorin Vasile Savu, Gabriel Constantin Benga, Laura-Madalina Cursaru, Dumitru Valentin Dragut, Roxana Mioara Piticescu, Danut Nicolae Tarniță

**Affiliations:** 1Department of Engineering and Management of Technological Systems, University of Craiova, 1 Calugareni Str., 220037 Drobeta-Turnu Severin, Romania; ionel.savu@edu.ucv.ro (I.D.S.); gabriel.benga@edu.ucv.ro (G.C.B.); 2Department of Applied Mechanics, University of Craiova, 107 Calea Bucuresti Str., 200512 Craiova, Romania; 3Advanced and Nanostructured Materials Laboratory, National R&D Institute for Non-Ferrous and Rare Metals, 102 Biruintei Bv., 077145 Pantelimon, Romania; mpopescu@imnr.ro; 4Analysis Laboratory, National R&D Institute for Non-Ferrous and Rare Metals, 102 Biruintei Bv., 077145 Pantelimon, Romania; dragutv@imnr.ro; 5National R&D Institute for Non-Ferrous and Rare Metals, 102 Biruintei Bv., 077145 Pantelimon, Romania; roxana.piticescu@imnr.ro; 6Department of Anatomy, Faculty of Medicine, University of Medicine and Pharmacy, 2 Petru Rareș Str., 200349 Craiova, Romania; dan_tarnita@yahoo.com

**Keywords:** polymer-shell powder composite, microwave heating, plasma discharge, thermal degradation, thermal field, Finite element analysis, numerical simulation of heat flow

## Abstract

During the microwave sintering of a polymer-ceramic composite plasma discharge is experienced. The discharge could occur failure of the power source. The solution proposed by the paper is original, no similar solutions being presented by the literature. It consists of using a polymer-ceramic composite protective panel, to stop the plasma discharge to the entrance of the guiding tunnel. Six composites resulted by combining three polymers, Polytetrafluoroethylene (PTFE), STRATITEX composite and Polyvinylchloride (PVC) with two natural ceramics containing calcium carbonate: Rapana Thomasiana (RT) sea-shells and beach sand were used to build the protective panel.Theoretical balance of the power to the panel was analysed and the thermal field was determined. It was applied heating using 0.6-1.2-1.8-2.4-3.0 kW microwave beam power. The panels were subjected to heating with and without material to be sintered. It was analyzed: RT chemical (CaCO_3_ as Calcite and Aragonite), burned area (range: 200–4000 mm^2^) and penetration (range: 1.6–5.5 mm), and thermal analysis of the burned areas comparing to the original data. PTFE-RT composite proved the lowest penetration to 0.6 and 1.2 kW. Other 1.2 kW all composites experienced vital failures. Transformation of the polymer matrix of composite consisted of slightly decreasing of the phase shifting temperature and of slightly increasing of the melting start and liquidus temperature.

## 1. Introduction

Specific application of polymers, as gears, or actuators, require improved mechanical behaviour of the material, as high mechanical resistance mixed with good wear resistance and/or high plastic behaviour. Nowadays, large number of polymers meet partially the required properties and the scientific research has an important direction to the development of different kind polymer-based composites. Such of a composite is a ceramic reinforced composite based on a common polymer, which is (PTFE) [1,2]. The ceramics used for reinforcement could be of many types, but a ceramic, which is simply available, is the sea-shell of (RT). (RT) is mainly composed of calcium carbonate (CaCO_3_) that is natural formed, and has high values for the mechanical properties. A PTFE-RT composite would combine the plasticity and the mechanical resistance and the electrical resistance of the PTFE matrix with the very high mechanical properties and high wear behaviour of the RT.

PTFE is one of the best-known commercial polymers, mainly due to higher dielectric properties and good chemical inertness [1,2]. According to Bur [3], its dielectric constant is small, equal to about 2, while the dielectric loss is very small, of the order of 10^−4^, giving a high stability over a wide domain of frequencies. Among the disadvantages of PTFE should be highlighted: a high linear coefficient of thermal expansion, low surface energy, low thermal conductivity and low dielectric constant [4]. Numerous researches have been undertaken, aiming at the adequate incorporation of the ceramic filler, in order to improve the mechanical and dielectric properties of PTFE [5,6,7,8,9].

In [10] by using micron and nano-ceramic fillers by hot pressing, (PTFE)–Mg_2_SiO_4_ composites have been prepared. Considering the dielectric properties of the composites as a function of reinforcer loading up to 50 vol%, they are investigated both at radio and microwave frequency ranges. The conclusions of the paper are that the dielectric constant and loss tangent increases with filler volume fraction

Marine organisms are natural materials such as corals [11], snails [12], shells [13], eggshells [14], and are important in terms of the CaCO_3_ content from which Hydroxyapatite (HA) can be obtained. Obtained from marine organisms, HA, which is classified as bioactive, osteo-conductive and biocompatible with hard tissues, has an important advantage, such as the possibility of its use in medicine, in bone tissue engineering, because it causes a better tissue response due to its similar properties. its chemical and structural properties with the inorganic constituents of biological hard tissues.

(RT), a predatory gastropod considered a threat to the environment [15], is a marine gastropod of Asian origin, which was brought accidentally with the ship ballast waters and it was observed for the first time in Romania in 1961. Since then, this species has established as a population in the Black Sea. It is an invasive species and it is commercially exploited in Romania since 2010s in gastronomic sector [16,17]. Its global availability is some of the main advantages that make it possible to use it in different applications [18]. It has a very high CaCO_3_ content (95–99% by weight), and mechanical properties similar to hard tissues such as human bones, low cost of production [18]. As result, the empty (RT) shells can be used as a natural source of calcium for the synthesis of biomaterials with applications in bone tissue engineering. They can be collected directly from the Black Sea coast or from fisheries [16,17].

Over time, several marine organisms (corals) aquatic species (fish bone, seashell, clam) and animal bones (bovine, pig, horse) were used as a source of calcium for hydroxyapatite production (as the main inorganic constituent of bone). Corals are used in the bone tissue engineering with reported clinical success and low complication rates. Stony corals produce an external calcium carbonate matrix with an open, highly interconnected porous structure and mechanical properties similar to that of bone. Calcium carbonate skeletons can be used as bone scaffolds either directly or converted into hydroxyapatite (non-resorbable) by hydrothermal exchange. An important drawback for coral bone substitutes is the lack of sufficient raw material. Their harvesting implies the destruction of coral reefs and therefore some species may be lost [19].

Tihan and colab. [20] presented a new method for the production of biomaterials based on natural polymers and marine organisms, in particular RT shells. Fourier transformation infrared spectroscopy (FT-IR) analysis and scanning electron microscopy (SEM) images of the obtained biomaterials confirmed the transformation of CaCO_3_ obtained from recycled RT shells into HA. Using FT-IR, the interactions between the biomaterial components were analyzed, and using SEM, the surface morphology was studied. CaCO_3_ is an extremely important material, both in basic research and in industrial applications, due to its beneficial properties, such as large surface area to volume ratio, non-toxicity and biocompatibility [21]. The global availability of CaCO_3_ and its important characteristics make the synthesis of this material a very attractive research topic for scientists. Biomimetic synthesis involves imitating nature’s ability to control the phase of (CaCO_3_), size and shape, based on the use of organic compounds, requires a small amount of additive, without changing the chemical properties of CaCO_3_ and provides a very large set of results due to a large number of organic additives. CaCO_3_ is strongly studied for its important role in the design of new composite materials [21].

Large number methods for HA extraction from animal bones used calcination, which involves heating the bone in a furnace at different temperatures of up to 1400 °C in order to completely remove the organic matter and kill the pathogens which may be present. Marine shells are rich with calcium carbonate (CaCO_3_) which can be converted to HA. However, several synthesis steps are required to produce high purity HA. The two main methods of preparing HA are wet chemical method (precipitation, hydrothermal, and the hydrolysis of other calcium phosphates) and solid-state reaction (sintering powders at high temperature). Vecchio et al. [22] were one of the first authors that reported the hydrothermal conversion of seashells (conch shell and clam shell) into hydroxyapatite, by mixing small pieces of these shells with diammonium hydrogen phosphate solution at 200 °C for several days. In 2011, S-C. Wu et al. [23] reported the synthesis of hydroxyapatite from grounded oyster shells mixed with synthetic calcium phosphates by milling (using ball milling process), followed by a specific heat treatment. Mocioiu et al. [24] have recently shown an innovative route to produce 3D scaffolds made of hydroxyapatite for further applications in bone tissue reconstruction. Thus, HA was prepared by hydrothermal synthesis in high pressure conditions, starting from RT as calcium source. HA powder was further used to fabricate 3D structures by extrusion-based 3D printing. Preliminary cytotoxicity tests performed on these 3D scaffolds are promising for biomedical applications.

Other methods for synthesizing CaCO_3_ particles include the microwave-assisted method, a better alternative to thermal heating [25]. The application of an alternating current through two cells containing Ca^2+^ and CO_3_^2−^ and separated by a (PTFE) membrane results in different supersaturation around the membrane [26]. The addition of CaCO_3_ particles improved the mechanical and rheological properties of different plastics such as poly (l-lactic acid) (PLLA) [21], PVC [27] polypropylene (PP) [28], and high-density polyethylene (HDPE) [29]. There are several very important advantages of the method of manufacturing hybrid polymer-inorganic materials and composites under microwave irradiation, such as: reduced processing time, more uniform heating of materials, faster curing of resins. In the case of polymeric hybrid materials, microwave-assisted synthesis implies advantages such as: smaller particle size, smaller particle size distribution, higher particle density, advantages that lead to the obvious improvement of the material characteristics obtained in the end. In [30] Sahebian and colab. presented results published in the literature on the preparation and characterization of composite materials and polymeric hybrids obtained by microwave irradiation using different types of polymer matrix and resins together with inorganic materials such as glass, carbon fiber, laminated materials. In [31,32] Bogdal and colab. present results on the synthesis of polymer-inorganic hybrid nanocomposites under microwave irradiation, a field of research with a rapid upward trend in recent years.

Authors searched for best-available technology to produce PTFE-RT composite. When try to elaborate the PTFE- RT composite by microwave melting strong polarization of the ceramic materials in microwave reaction chamber is experienced. Such polarization usually led to plasma discharge (Figure 1) from ceramic composite sample towards magnetron antenna. Failures of the microwave equipment can occur as result.

That would be explained by electrical reasons. Microwave heating of materials is a process with high rate of heating, based on several heating mechanisms, the most important being the dipole rotation phenomenon. The electromagnetic waves with high frequencies [33] have wavelengths ranging from 1 m to 1 mm being indirect proportionally with frequency. The dipole rotation heating mechanism consists of frictions between dipoles inside materials that change their orientation as function of changing direction of electrical field lines. Not all materials are susceptible to convert microwaves into heat due to their low value of the loss tangent represented by ratio between loss factor and dielectric constant property. Polymers have low dielectric constant (PTFE = 2.1, LDPE/HDPE = 2.25, PVC = 3, etc.) and they are almost transparent to microwaves. On the other hand, ceramics have high losses in medium and therefore high rate of conversion in heat. Due to all these, plasma discharge is unstable and randomly oriented, but mainly directed to the magnetron producing over-heating of it. A solution against this phenomenon would be the insertion into the waves guiding tunnel of a protective panel which is transparent for microwave beam and has the ability to stop the microwave plasma discharge. Such panel should have the following characteristics: a. to be transparent for the microwave beam with low and mild power (up to 3 kW) in order to not develop heating process due to the interaction with the microwave beam, and b. to have the ability to oppose and block the discharge of plasma created by the microwave beam due to the high-level ionization of the gas in the processing chamber.

Transparent for microwaves are most of the polymer types. No polymer can perform an appropriate function as protective panel due to its critical temperatures (glassy and melting and thermal degradation) which are lower [34,35] than the temperatures produced and radiated from the heating chamber the temperatures of the microwave plasma produced in the heating chamber.

The solution is to prepare a composite polymer-ceramics, in which the ceramics to be a thin layer positioned on the side attacked by the plasma discharge. Since the material subjected to sintering is the powder produced by milling pieces of RT shells, the first ceramic material proposed for the building of the heat protective layer is the same RT powder. Taking account that the shells and the beach sand contain mainly CaCO_3_ and SiO_2_, the beach sand is proposed as second potential material for the building of the heat protective layer.

The paper presents the research on the behavior of the 6 composites (PTFE-RT, PTFE-sand, STRATITEX-RT, STRATITEX-sand, PVC-RT, and PVC-sand) as protective panels in the process of microwave heating of the PTFE-RT composite.

State-of-the-art literature, in the field of microwave heating, does not present scientific solutions for protection of the microwave generator antenna. Most of the researches were focused on hybrid microwave heating [10,33] meaning that the samples were introduced in most of the cases into SiC crucibles. Thus, the microwave processing was an indirect one, the samples being heated by thermal radiation accumulated from the silicon carbide crucibles. It is well known that silicon carbide is a very good microwave to heat converter, while providing process stability [29,30]. However, the adoption of this technology does not ensure pure microwave heating, which leads to the growth of grains in the material, even if the heating process is very fast. The authors’ research aimed at direct microwave heating of calcium carbonate samples. Thus, the probability of the appearance of the microwave electric arc is very high, and the novelty of the technology consists in approaching a pure microwave heating while ensuring a protection of the generator antenna by implementing protective panels of polymers with high dielectric strength that are resistant to high temperatures. No similar solution was proposed by the literature. All authors preferred to apply lower power of the microwave beams, or to use controlled environment, in order to reduce the effect of the discharge [6,7,31,32], instead blocking the discharge that always is experienced, cracking the sintered piece and affecting the magnetron. The composite structure of the panel is, also, an original solution, using natural ceramics that are waste. Defining a range of functionality for the protective panel is novelty, as well.

## 2. Materials and Methods

As already specified, the protective panel is proposed to be inserted into the microwave guide tunnel, in order to create a barrier against the plasma produced by the ionizing of the gas in the oven, during the interaction between the microwave beam and the piece to heat or between the microwave beam and the walls of the oven. The protective panel is transverse positioned on the wave guide tunnel and its dimensions were considered according to the geometry and structure of the microwave guiding tunnel connected to the matching load impedance auto-tuner TRISTAN 6 kW (Figure 2) [36] driven by HOMER software (producer of the entire system is MUEGGE GmbH, Reichelsheim, Germany). The heating environment was simple air, and the room temperature was 22 °C for all experimental steps.

The materials to be heat was mix of PTFE grains (max 500 µm thick) and RT powder with a granulation of 1500–2500 nm (Figure 3). The mass ratio of mix was 70% weight mass PTFE and 30% weight mass RT powder, for each test. The RT powder was milled within Pulverisette 6 planetary mill [37], for 2 h at 400 rot/min speed, starting from chips of sea shells up to 5 mm dimensions.

The chemical composition of RT powder was investigated by inductively coupled plasma-optical emission spectrometry (ICP-OES) using Agilent 725 ICP-OES instrument (Agilent Technologies, Santa Clara, CA, USA) [38]. Ca, Na and K elements were determined by atomic absorption spectrometry performed with Analytik Jena ZEEnit 700 AAS Atomic Absorption Spectrometer (Jena, Germany) [39].

The crystalline phases present in the RT powder were identified by X-ray diffraction (XRD) using Bruker D8 ADVANCE diffractometer (Billerica, MA, USA) [40]. The data processing was done with the help of the DIFFRAC.EVA VER.5 2019 program and the ICDD database PDF4 + 2021 [41].

For the protective panels used in the experimental program, two polymeric materials and a composite were subjected to testing: PTFE, STRATITEX (which is laminated board based on cotton fabric and phenolic resin) and PVC (Figure 4). All panels were 150 × 100 mm^2^, and 10 mm thick. They all are transparent for the microwaves and very low interaction between them and the microwave beam can be recorded. In opposition, they are interacting with the microwave plasma discharge, which is in the range of 100–300 °C when the input power increases from 600 W to 3000 W. That is the reason of using specific layer resistant to the plasma impact, and the ceramics are the most recommended here. In the same time, the ceramics are interacting with the microwave beams and heat will be generated to the surface of the protective panel.

As result, on each protective panel thin layers of RT powder or sand was deposited (Figure 5). The depositions of the layers were done using the same method. The RT powder, having a granulation of 1500–2500 nm, is mixed with copolymer granules and vinyl acetate. The mixture is placed on the polymer protection panel. They are heated by radiation from a resistive source up to 180 °C. At this temperature the copolymer melts, while the polymeric support is in a fluid-viscous state, and the deposition layer is created by combining the two.

The obtained composite is a polymeric one with natural ceramics reinforcement. The polymeric support allows the microwave beam to pass without interaction and the ceramic surface acts as thermal barrier to the plasma discharge.

Each sample of protective panels was installed into the structure of the microwave heating equipment, in specific place between the wave guidance tunnel and the heating chamber, as specified above (Figure 6). The installing was done by mounting screws. During any heating process the incident wave passed through the protective panel as an energy beam, and hit the mixture of polymer and RT powder, existent in a small ceramic crucible. The type of mixture (mass percentage) and the quantity of mixture were maintained as constant parameter from test to test.

For the heating, a 3 kW (50% of max), 2.4 kW (40% of max), 1.8 kW (30% of max), 1.2 kW (20% of max), and 0.6 kW (10% of max) power has been adjusted to the equipment.

At such levels of power (usual for sintering processes) plasma discharge in air environment is formed when use ceramics or ceramic based composites or ceramic reinforced composites probe in the oven. By decreasing the power, the stiffness and the temperature of the plasma discharge is decreasing, consequently, and the protective panel is less and less affected by the thermal shock produced by the plasma discharge.

## 3. Theoretical Approach

The experimental setup composed of: magnetron to produce microwaves, microwave orientation tunnel, load impedance auto-tuner, protective panel and oven is an enhanced setup used to apply the microwave heating processing. Such setup produces an energy balance according to Figure 7. The input energy (E_1_) injected by the magnetron passes through the polymeric component of the composite that creates the protective panel. When touches the ceramic layer (RT or sand) interaction between the two produces an amount of heat (E_4_), which has and increasing of speed from the inside of the ceramic layer to the outside. E_4_ is able to heat the entire protective panel by conduction and that is a negative effect created by the ceramic reinforcement of the composite to the protective panel.

The passed amount of energy touches the PTFE-RT probe and the walls of the oven. First, a reflected wave will be produced and that wave will carry E3 amount of energy back to the magnetron. This amount of energy can be partially transformed of microwaves plasma if the environment permits ionization of its atoms. After passing the protective panel E_3_ is decreased by the load impedance auto-tuner. A second amount of energy returns from the oven to the protective panel and that is E_2_, which is a heat radiation from the PTFE-RT probe. Partially, E_2_ participates to the ionization of the environment and to the creation of the plasma discharge. The plasma discharge produced by both together, E_2_ and E_3_, should be stopped by the protective panel from its evolution to the magnetron.

In brief, E1 and E3 produce heat within the ceramic layer by the interaction between the microwave beam and the ceramic material: $Q_{1}$ and $Q_{3}$. E_2_ brings heat to the surface of the ceramic layer: $Q_{2}$. The energy balance on the ceramic layer becomes:(1)$$Q_{rec\_cer}=Q_{1}+Q_{2}+Q_{3}$$
where $Q_{rec\_cer}$ is the quantity of heat received by the ceramic layer of the protective panel, $Q_{1}$ is the quantity of heat produced by the interaction between the incidental microwave beams and the ceramic layer of the protective panel, $Q_{2}$ is the quantity of heat radiated from the sample to heat, and $Q_{3}$ is the quantity of heat produced by the interaction between the reflected microwave beams and the ceramic layer of the protective panel.

In the same time, the polymeric panel is subject to heat incidence coming by conduction from the ceramic layer of the composite, only. The polymer does not absorb microwaves, so the energetic amounts E_1_ and E_3_ refer to the ceramic layer and do not affect the polymer part of the composite.
(2)$$Q_{rec\_polym}=Q_{4}$$

In Equation (2) $Q_{rec\_polym}$ is the quantity of heat received by the polymer layer of the composite panel, and $Q_{4}$ is the heat which is transferred by conduction from the ceramic layer to the polymer layer of the composite protective panel.

$Q_{1}$ and $Q_{3}$ have similar structure, even if the differences between the transferred energy is high. E_1_ is the incident beam of microwaves and depends on the input energy by the magnetron. $Q_{3}$ is the reflected energy, meaning the amount of energy reflected by the piece subjected to heating, and by the heating chamber-walls. $Q_{3}$ is measured to be around 15–25% of the incident energy. $Q_{4}$ is the transferred amount of heat, by conduction, from the ceramic layer to the polymeric support, and depends on the $Q_{rec\_cer}$ quantity of heat produced by E_1_ + E_2_ + E_3_ within the ceramic layer.

To have an appropriate analysis of the heat flowing from the ceramic layer to the polymer it has been considered the entire thickness as being composed of 6 elements of 2 mm. The reason is that after the initiation of the heating by conduction the polymer is not totally transparent anymore and experiences heat development due to the interaction with the incident microwave beam. The first element of the polymeric support, in direct contact to the ceramic layer and containing RT or sand powder due to the elaboration technique reaches after ∆θ time a temperature of:(3)$$T_{x}^{,}=\frac{∆\theta }{c\cdot \rho }\cdot \left[\frac{q}{\delta x}+\frac{k_{R}\left(T_{R}-T_{x}\right)}{\delta x^{2}}\right]+T_{x}$$
(4)$$∆\theta =\frac{c\cdot \rho \cdot \delta x^{2}}{k\cdot Mat}$$
(5)$$Mat=\frac{c\cdot \rho \cdot \delta x^{2}}{k\cdot ∆\theta }=\frac{\delta x^{2}}{\alpha \cdot ∆\theta }$$
where Mat is a parameter that describes the evolution of the material along the axe of the heat flowing (δx). The next elements reach the temperatures of:(6)$$T_{x}^{,}=\frac{∆\theta }{c\cdot \rho \cdot \delta x^{2}}\cdot \left[k_{L}\left(T_{L}-T_{x}\right)+k_{R}\left(T_{R}-T_{x}\right)\right]+T_{x}$$

E_1_ and subsequently, but indirect, $Q_{1}$ is correlated with the evolution of the electric field. The electric field is perpendicularly on the magnetic field as a uniform plane wave, depending on the electric susceptibility. The evolution of the microwave beam is in single direction, and this produces a one-dimensional transfer of heat.

A particular case of the heat balance is the situation when no probe to heat exists. In that case E_2_ is heading to zero and E_3_ is increasing. Regarding E_3_ evolution, a higher amount of energy is reflected from the walls, instead the general situation, with a heating probe, when part of the energy is absorbed by the probe.
(7)$$Q_{rec\_cer}=Q_{1}+Q_{3}$$

A second particular case is the situation when the protective panel is bult of polymer only, without ceramic shielding against plasma discharge. It is the simplest version of protective panel and, due to the transparency to microwave beams of polymers, no heat sources related to interaction between microwaves and materials will occur. In that case E_1_ and E_3_ are heading to zero and the single heating component is E_2_, producing $Q_{2}$.
(8)$$Q_{rec\_cer}=Q_{2}$$

The specific thing here is that the panel is facing plasma discharge attack and, even if the plasma is a “cold” one, below 300 °C temperatures being measured in previous researches, the polymer does not resist and enters the fluid-viscous state and even experiences local thermal degradation.

In any case, the heat transfer by conduction from the ceramic layer to the polymeric support can be accepted as:(9)$$\rho \cdot C_{p}\cdot \frac{\delta T}{\delta t}-\nabla \cdot \left(k\nabla T\right)=Q$$
where ρ is the density of the polymers (2.2 g/cm^3^ for PTFE, around 1.78 g/cm^3^ for STRATITEX, and 1.38 g/cm^3^ for PVC), $C_{p}$ is the specific heat at constant pressure ($C_{p}$ = 970 J/KgK for PTFE and is 880 J/KgK for PVC), and k is the thermal conductivity (k = 0.25 W/mK for PTFE and between 0.12–0.25 W/mK for PVC). The boundary condition for the heat flux inside the polymer is:(10)$$-n\cdot \left(-k\nabla T\right)=h\cdot \left(T_{amb}-T_{max}\right)$$
in which $T_{amb}$ = 22 °C is the minimum temperature (Ambiental) and $T_{max}$ = 300 °C represents the maximum temperature measured (using IR pyrometer) on the surface of the ceramic layer.

Applying numerical simulation to the mathematical model of the thermal field it can be observed (Figure 8) the following behavior of the composite protective panel.

The values were confirmed by Finite element analysis (FEA) simulation of the model (Figure 9).

It can be observed in Figure 9 that the penetration of the heat is about 4 mm for temperatures above 250 °C, about 6 mm for temperatures above 200 °C, and about 8 mm for temperatures above 130 °C. The side of the panel oriented to the magnetron keeps to be at temperatures below 50 °C. That means that the protective panel has appropriate behavior.

## 4. Results and Discussions

The chemical composition of RT powder is presented in Table 1. It can be observed that (RT) shells are an important source of calcium. Small amounts of Na, Sr, Mg, S and Si are also identified. Qualitative phase analysis of RT powder by X-ray diffraction (XRD) is shown in Figure 10. Crystalline phases identified in (RT) shells are highlighted in Table 2.

As it can be seen in Table 2, (RT) shell is made of calcium carbonate crystallized as calcite (~64.5%) and aragonite (~35.6%).

During preliminary tests, experience of burning with open flame and emissions of fume (Figure 11) proved that the power was appropriate for testing, due to the existence of the plasma discharge. For 1.2 kW and 0.6 kW input power the plasma discharge did not performed outside evolution, and the amount of fume emission was almost not existent. The two levels of power proved to be more than enough to melt the PTFE from the probe and to produce the desired composite.

For higher amounts of power, the process was difficult to be controlled and the penetration of the protective panel was almost on the entire thickness. The images (a), (b), and (c) of Figure 12 show the power curves (incident, reflected and absorbed) for the three types of protective panels using RT layer, and for 3 kW input power. In less than 20 s the plasma discharge appeared and hit the protective panel. Image (a) of Figure 12 shows that in the case of PTFE the process has higher stability. The images (d), (e), and (f) show the Schimdt curves of stability. For PTFE case the process is very stable, since the curves are in the center of the circle, which is different from the STRATITEX case when the process is out of control. In the case of STRATITEX, TRISTAN impedance matcher is almost unable to adjust the stubs in order to reduce the reflected beam.

All samples were burned by the developed heat, each with own specific intensity, according to the energy balance presented above, and according to the heat characteristics of the materials involved (Figure 13).

It can be observed (especially in Figure 13d) that the burning area has the shape given by the sinusoidal evolution of the electrical field, and over the burned area due to the interaction between the microwaves beam and the panel burning given by the plasma discharge occur. The area burned by the plasma discharge has no particular shape, because the discharge is very dynamic as position. Even if the stiffness of the discharge increases with the input power, the area of the burned surface keeps to be high due to the increasing of the plasma temperature. It can be, also, observed a difference of about 15–20% difference between the areas of the burned surfaces when used RT layer or sand layer. When use RT powder to build the layer the area of the burned surface is lower. Regarding the penetration into the panel, in all tests the penetration when used sand was higher with up to 50% than the situation when RT layer was used. A lower input power produced a higher difference between the penetration values.

For the lowest two levels of input power (1.2 kW and 0.6 kW) the entire heating process was very stable and the protective panels were affected by penetration of 0.0–1.0 mm, and respectively 0.0–0.6 mm (Figure 14).

Important amount of energy, participating to the plasma discharge, comes from the material subjected to heating. That influences the interaction between the plasma and the protective panel, and it is considered here the penetration of the panel. Since the RT layer proved the best behavior during the interaction with the plasma and with the incident microwave beam, an experiment without sample to be heat was performed. That was necessary to better understand all the influences. The burned samples are presented in Figure 15a–c and it can be seen that the penetration is max 10% lower than the previous tests, when sample of PTFE-RT was positioned in the oven.

In conclusion, the composite PTFE-RT proves the best response to the heating process.

It was expected that the burning of the panel to return modifications of the polymer and the plasticity of the burned PTFE (polymer that proved the most appropriate evolution as support for the protective panel) was evaluated by applying thermal analysis—Differential Scanning Calorimetry (DSC). The DSC determinations were done according to ASTM D3418 and they were accomplished using specific calorimeter NETZSCH [42], type 204 as follows: in inert atmosphere, heating from 22 °C to 210 °C, with a rate of 10 °C/min, cooling at −100 °C with a rate of 10 °C/min, isothermal regime at −100 °C for 5 min, heating at 400 °C with a rate of 5 °C/min. The test was used as tool to reveal the changes in specific heat capacity of the material positioned on the surface of the protective panel against the material positioned on the bottom of the heated area, so the most affected area during the heating process. The result of the DSC analysis is presented in Figure 16a,b. It can be observed that the phase transition takes place at almost the same value of temperature (Figure 16a), which is different from the melting process, in which the differences are more sensitive (Figure 16a,b). Figure 16b shows the melting behavior (liquid content, liquid fraction) and the recorded curves show sharp melting (the enthalpy of transformation is related to the area under the peak). There was recorded a difference of more than 4 mJ between the peaks’ areas, the value for the bottom of heat affected area being higher. That means that the burned material modified its capacity to melt, becoming more difficult to be melted. In the same time, the endothermic melting, given by the ranges between the melting start temperature and the liquidus temperature moved to higher temperatures, from (old position: 276.25–346.80 °C, and the new position: 278.46–347.76 °C), according to Figure 16b. The translation of the melting range proves again the modifications suffered by the PTFE during the heating process.

## 5. Conclusions

The paper presented a solution to block the plasma discharge created when apply sintering process to different types of composites. Even if the phenomenon was experienced by the most of the researchers using microwaves for heating, no solution to block the plasma discharge to not affect the magnetron. The researchers were oriented to change the environment in which the heating to be applied, measure that is, generally, expensive.

The proposed solution is to use protective panel built of polymer-ceramic composite. The panel should be as transparent as possible for the microwave beams, but it should be able to resist to the plasma discharge, as well. The composite consisting of polymer support (transparent to the microwaves) and a heat-resistant layer produced of natural ceramics, extracted from waste (able to facing the impact with the plasma discharge) gives appropriate response, being a cheap and effective material.

Such proposed protective panel inserted into the microwave guiding tunnel is an original solution to reduce the risk of failure or even total burning of the magnetron.

Within the paper it was determined the energy balance at the panel level by own mathematical model and the calculated data were confirmed by FEM analysis. The confirmed data was the base of the heating parameters’ values selection. The selected values were applied to 6 composites (combinations of 3 polymers and 2 natural ceramics resulted from waste)

The most appropriate results (in terms of penetration depth and burned area) was recorded for the composite PTFE-RT.

The input power proved important influence on the burning process, on the plasma discharge characteristics, and on the burned panel material, and all 6 combinations of composites returned appropriate behavior for power values below 1.2 kW.

The proposed solution protects from direct burning and heating both the magnetron and the automatic adjusting system of the guiding tunnel for the microwave equipment. That is converted in thousands of EUR saved. Replacement of ceramic layer with lower microwave absorption material is task for future research, in order to increase the life of the protective shield and the input power from max 1.2 kW to 6.0 kW.

## Figures and Tables

**Figure 1 polymers-13-02432-f001:**
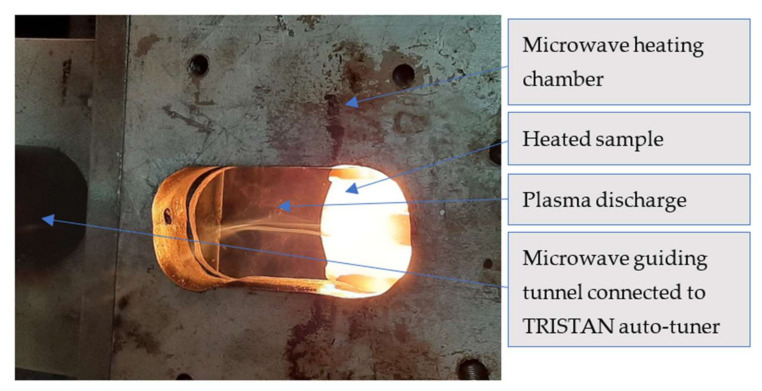
Plasma discharge from the heated sample to the wave guiding tunnel.

**Figure 2 polymers-13-02432-f002:**
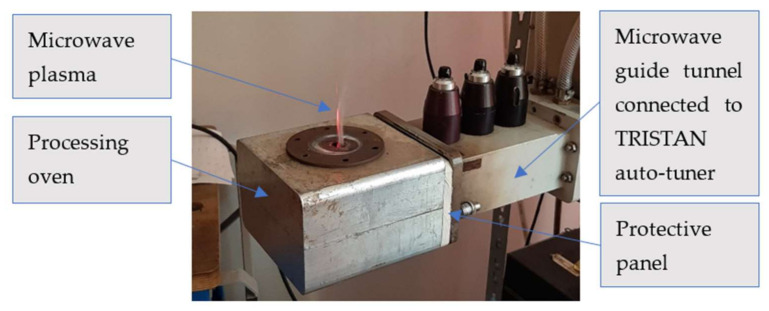
Position of the protective panel within the microwave machine structure.

**Figure 3 polymers-13-02432-f003:**
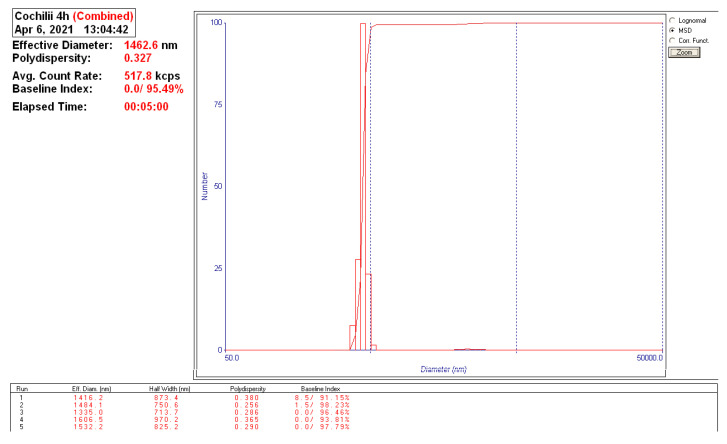
Grain size distribution of the RT powder (Malvern Nanosizer).

**Figure 4 polymers-13-02432-f004:**
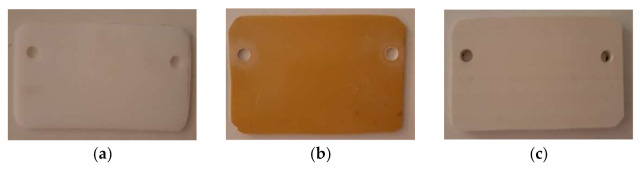
Samples prepared for heating: (**a**) PTFE; (**b**) STRATITEX; (**c**) PVC.

**Figure 5 polymers-13-02432-f005:**
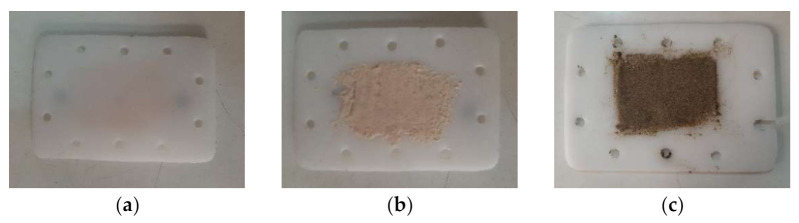
PTFE samples prepared for heating: (**a**) simple; (**b**) with RT powder layer; (**c**) with sand layer.

**Figure 6 polymers-13-02432-f006:**
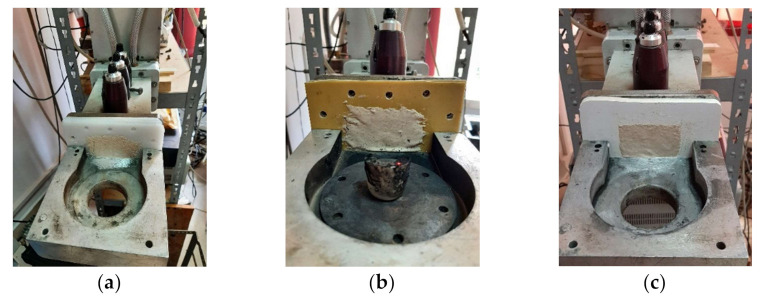
Samples prepared for heating: (**a**) PTFE; (**b**) STRATITEX; (**c**) PVC.

**Figure 7 polymers-13-02432-f007:**
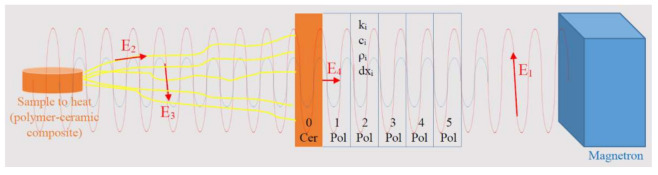
Energy balance.

**Figure 8 polymers-13-02432-f008:**
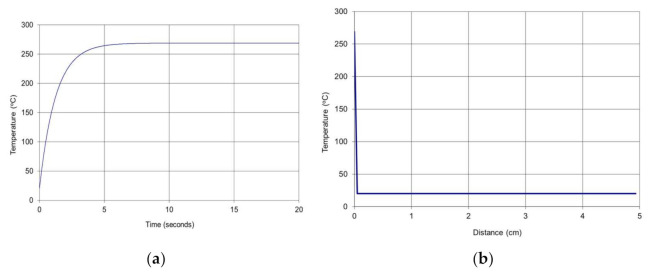
Numerical results: (**a**) Transient temperature in the composite; (**b**) Temperature versus X direction of heat evolution.

**Figure 9 polymers-13-02432-f009:**
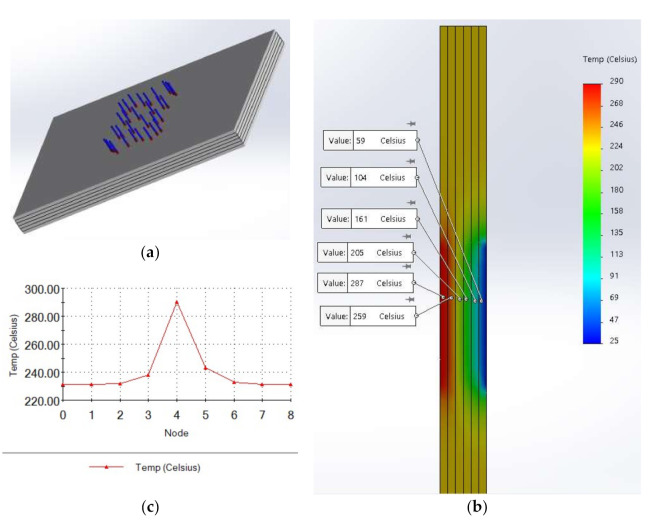
FEA simulation of the geometrical model: (**a**) Geometrical model of the composite; (**b**) Thermal field on thickness; (**c**) Temperature plots; (**d**) Thermal field on surfaces of each element of thickness considered.

**Figure 10 polymers-13-02432-f010:**
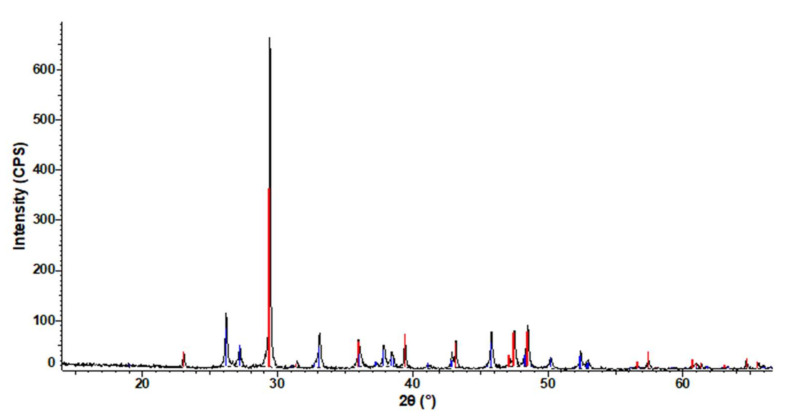
Graphical presentation of the qualitative phase analysis by XRD for RT powder. Main peak depicted in red corresponds to calcite, while secondary peaks marked in blue are assigned to aragonite phase.

**Figure 11 polymers-13-02432-f011:**
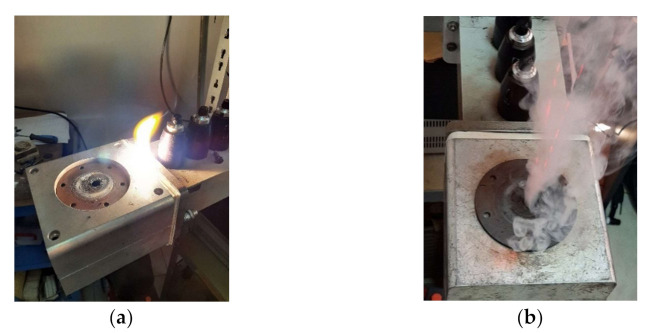
Images from the heating process: (**a**) microwave plasma discharge produced the burning of the sample; (**b**) Fume produced by the burning of the sample.

**Figure 12 polymers-13-02432-f012:**
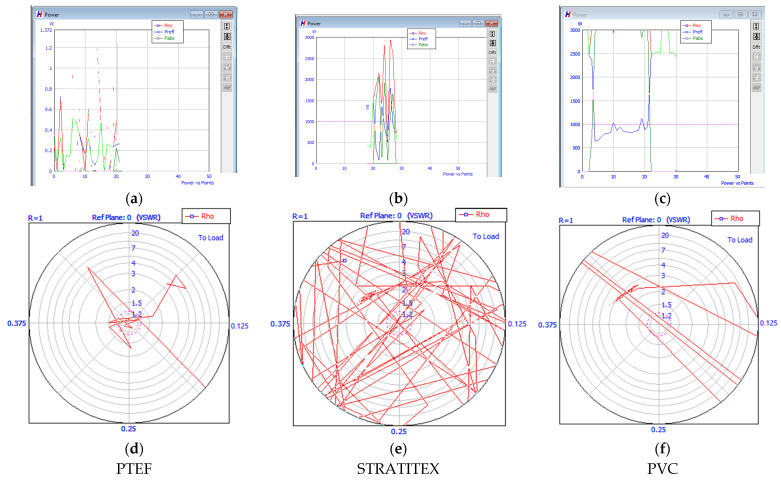
Heating’s characteristics records: (**a**–**c**) Recorded waves, in W—incident/red, reflected/blue, and absorbed/green; (**d**–**f**) Schmidt curves of stability.

**Figure 13 polymers-13-02432-f013:**
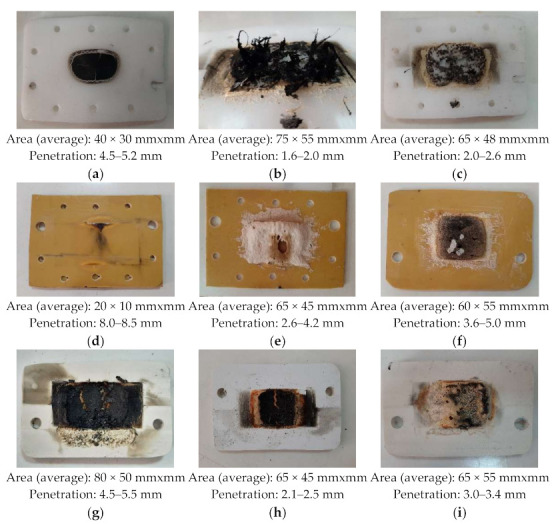
Examples of burned samples, input power 3 kW, burn time 20 s, PTFE+RT mixture to melt: (**a**) PTFE, no ceramic layer; (**b**) PTFE, RT layer; (**c**) PTFE, beach sand layer; (**d**) STRATITEX, no ceramic layer; (**e**) STRATITEX, RT layer; (**f**) STRATITEX, beach sand layer; (**g**) PVC, no ceramic layer; (**h**) PVC, RT layer; (**i**) PVC, beach sand layer.

**Figure 14 polymers-13-02432-f014:**
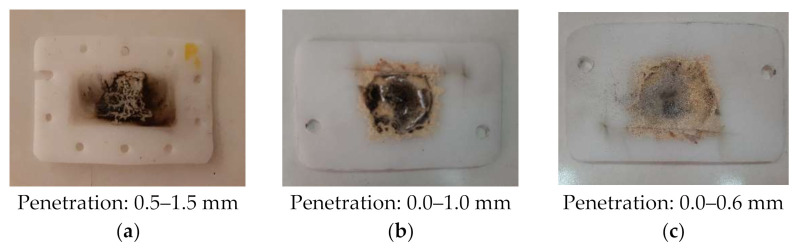
Examples of burned PTFE samples, input power: (**a**) 1.8-kW; (**b**) P1.2-kW; (**c**) 0.6-kW.

**Figure 15 polymers-13-02432-f015:**
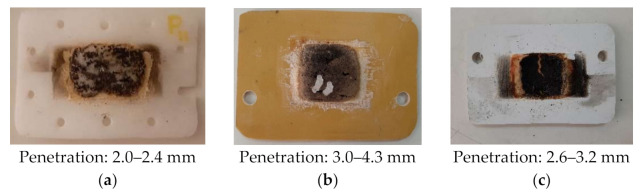
Examples of burned samples, input power 3 kW, burn time 20 s, without piece to melt: (**a**) PTFE, RT layer; (**b**) STRATITEX, RT layer; (**c**) PVC, RT layer.

**Figure 16 polymers-13-02432-f016:**
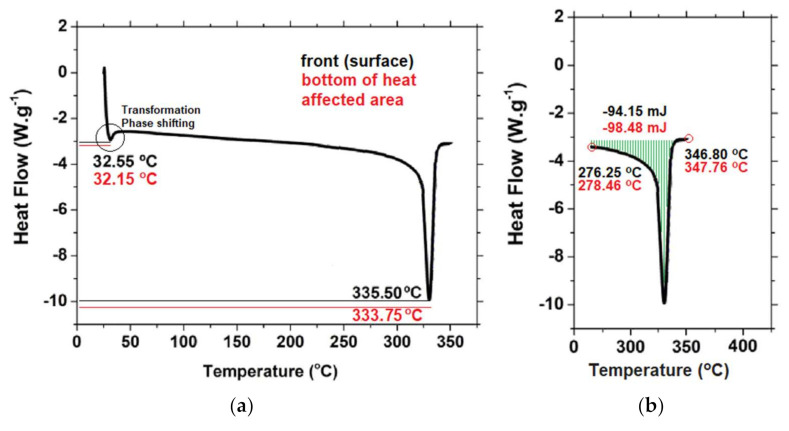
DSC analysis of the protective panel’s material (black-material on the surface of the affected side, and red—material on the bottom of the affected, on the same side): (**a**) Critical temperatures: melting temperature and phase transition temperature; (**b**) and the specific temperatures: melting start and liquidus temperature.

**Table 1 polymers-13-02432-t001:** Chemical composition of RT powder.

Sample Name	Unit	Al	As	Ba	Bi	Cd
(RT) powder	%	<0.005	<0.005	<0.005		<0.005
		**Co**	**Cr**	**Cu**	**Fe**	**Ga**
	%	<0.005	<0.005	<0.005		<0.005
		**Li**	**Mg**	**Mn**	**Mo**	**Ni**
	%	<0.005	0.072	<0.005	<0.005	<0.005
		**P**	**Pb**	**S**	**Sb**	**Si**
	%	<0.005	<0.005	0.038	<0.005	0.024
		**Sn**	**Sr**	**Ti**	**V**	**Zr**
	%	<0.005	0.11	<0.005	<0.005	<0.005
		**Zn**		**Na**	**K**	**Ca**
	%	<0.005		0.21	< 0.001	40.1

**Table 2 polymers-13-02432-t002:** Crystalline phases identified by XRD.

**Compound Name**	**PDF Reference**	**Chemical Formula**	**Crystallization System**	**Space Group**	**S-Q (%)**
Calcite	01-083-3288	CaCO_3_	Rhombohedral	R-3c (167)	~64.5
Aragonite	01-075-9982	CaCO_3_	Orthorhombic	Pmcn (62)	~35.6

## Data Availability

Not applicable. The results of the study were not published in other circumstances (journals, conferences, public databases, etc.).

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
