# Peer review of "Composite Polymer for Hybrid Activity Protective Panel in Microwave Generation of Composite Polytetrafluoroethylene -Rapana Thomasiana"

_polymers, 2021, doi:10.3390/polym13152432_

Round 1

Reviewer 1 Report

The manuscript is good. 

  1. The whole abstract needs to be rewritten. The significance and purpose of this research should be clearly presented in the abstract. The abstract must be presented in a clear way in problematic, objective, idea, description of idea, highlighting the methods, results, quantitative comparison of results with significant findings, conclusions.
  2. The state-of-the-art comparisons for the proposed work are missing in this paper. Then do a critical analysis of previous research. State explicitly the shortcomings of previous research. What is positive in previous research and what is negative. Based on that, you explicitly define the goal of the research and the scientific hypothesis.
  3. Highlight the novelty of your methodology.
  4. The Conclusion section should be rewritten. Highlight your scientific contribution. Highlight the benefits of your research. Define shortcomings and future research.

Reviewer 2 Report

The present manuscript investigates on the behavior of the 6 composites (PTFE-RT, PTFE- sand, STRATITEX-RT, STRATITEX-sand, PVC-RT, and PVC-sand) as protective panels in the process of microwave heating of the PTFE-RT composite. • The paper is well written and well organized. • The literature review is adequate however, some recently references can be added to the paper. The paper should be double check to avoid typo mistakes e.g. figure 10 should be labeled as a and b not c and d. • The methodology is presented in an acceptable way. It is interesting that the heat loading /unloading diagram at least as a result of FE simulation can be presented. • The results are good and they show interesting findings. • The conclusion is good and at the end after minor correction the paper can be published.
